# Collect & Infer - a fresh look at data-efficient Reinforcement Learning

**Martin Riedmiller**
DeepMind, UK

**Jost Tobias Springenberg**
DeepMind, UK

**Roland Hafner**
DeepMind, UK

**Nicolas Heess**
DeepMind, UK

**Abstract:** This position paper proposes a fresh look at Reinforcement Learning (RL) from the perspective of data-efficiency. RL has gone through three major stages: pure on-line RL where every data-point is considered only once, RL with a replay buffer where additional learning is done on a portion of the experience, and finally transition memory based RL, where, conceptually, all transitions are stored, and flexibly re-used in every update step. While inferring knowledge from all stored experience has led to a tremendous gain in data-efficiency, the question of how this data is collected has been vastly understudied. We argue that data-efficiency can only be achieved through careful consideration of both aspects. We propose to make this insight explicit via a paradigm that we call 'Collect and Infer', which explicitly models RL as two separate but interconnected processes, concerned with data collection and knowledge inference respectively.

## 1    Introduction

Data-efficiency in Reinforcement Learning (RL) can be loosely characterized as 'getting the most out of the collected experience'. Data-efficiency is critical in many real-world scenarios [1], where gathering data is the main bottleneck (e.g. in robotics), but it is also, arguably, a key property of Artificial General Intelligence (AGI). From a data-efficiency perspective, reinforcement learning methods have gone through three major stages. The original RL framework was phrased in a pure 'online' setting: the agent acts, observes the reward and new state, updates its behaviour and acts again. This view continues to be successful in settings where data is cheap, e.g. if a simulator of the environment is available. The next stage was to introduce a replay buffer [2], which stored a subset of the transitions to enhance the learning signal by iterating over recent experience multiple times. Building on previous work [3, 4], Ernst et al. [5] and Riedmiller [6] independently suggested to take this idea to the extreme, store all experience in a transition memory and re-use the

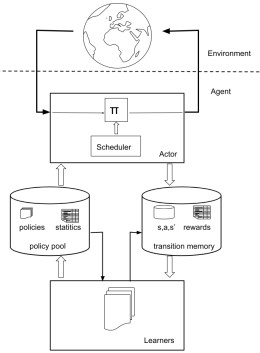

Figure 1: Collect and Infer Agent. Collect (upper part) and Infer (lower part) interact through a shared policy pool and transition memory.

full data in every update step. This led to a breakthrough in data-efficiency and made the application of model-free RL in the real world possible [7, 8]. Recent years have witnessed a revival of this idea with off-policy actor-critic algorithms rapidly gaining importance [9, 10, 11, 12]. In parallel there has been a growing interest in RL algorithms that can learn from fixed data sets entirely without interaction (*offline RL*) [13, 14, 15, 16, 17, 18, 19].

We extrapolate from these developments and argue that a clear conceptual separation of the reinforcement learning process into two distinct sub-processes, data-collection and inference of knowledge, will lead to further improvements in data efficiency and enhanced capabilities for the next generation of RL agents. We refer to this perspective as the *Collect and Infer* (*C&I*) paradigm. It assumes two sub-processes: acting (data *collection*), and learning (*inference*) which are decoupled but connected through a transition memory into which all data resulting from environment interaction is collected, and from which data is drawn for learning. A particular emphasis is put on how the data is collected. This view of RL as two independent processes provides additional flexibility in algorithm design and emphasizes that these processes can and should be optimized independently.

Blue Sky Papers, 5th Conference on Robot Learning (CoRL 2021), London, UK.

This paper gives a light-weight overview of the core concepts and implications of the *C&I* paradigm. We discuss recent examples from the literature, how these algorithms can be interpreted from the *C&I* perspective, and where that perspective suggests changes or improvements. We conclude with a discussion of research questions motivated by the paradigm.

## 2    The Collect and Infer paradigm and its implications

The key idea of the *C&I* paradigm is to separate Reinforcement Learning into two distinct but interconnected processes: process 1 deals with collecting data into a transition memory by interacting with the environment, process 2 infers knowledge about the environment by learning from the data of said memory. This perspective provides us with a new handle on the question of data efficiency which we can optimize by considering each process separately via the following objectives:

1. (O1) Given a fixed batch of data, what is the right learning setup, to get to the maximally performing policy (optimal 'inference')?

2. (O2) Given an 'inference' process, what is the minimal set of data, to get to a maximally performing policy (optimal 'collection')?

The *C&I* perspective has several implications. While it does not prescribe a particular algorithmic solution it encourages us to develop algorithms that satisfy the following desiderata:

1. Learning is done offline in a 'batch' setting assuming fixed data as suggested by O1. Data may have been collected by a behavior policy different from the one that is the learning target [e.g. 20]. This enables utilization of the same data to optimize for multiple objectives simultaneously, and coincides with interest in offline RL [13, 16, 18, 15, 19].

2. Data-collection is a process that should be optimized in its own right. Naive exploration schemes that employ simple random perturbations of a task policy, such as epsilon greedy, are likely to be inadequate. The behavior that is optimal for data collection in the sense of O2 may be quite different from the optimal behavior for a task of interest.

3. Treating data-collection as a separate process offers novel ways to integrate known methods like skills, model-based approaches, or innovative exploration schemes into the learning process without biasing the final task solution.

4. Data collection may happen concurrently with inference (in which case the two processes actively influence each other and we get close to online RL) or can be conducted separately.

5. *C&I* suggests a different focus for evaluation: in contrast to usual regret-based frameworks for exploration, *C&I* does not aim to optimize task performance during collection. Instead, we distinguish between a learning phase, during which a certain amount of data is collected, and a deployment phase, during which the performance of the agent is assessed.

Collect and Infer has implications for agent architectures, and it suggests alternative solutions to a number of problems that will become prominent as RL is applied to more challenging scenarios, including multi-task, transfer or life-long learning [21]. The Scheduled Auxiliary Control (SAC-X) architecture of Riedmiller et al. [20] exemplifies several of the above ideas. Its components are an actor, a transition memory, one or more learners, a pool of candidate policies and a scheduler, that selects policies for execution by the actor such as to collect experience that is informative for learning one or multiple tasks. Although the SAC-X agent does not explicitly optimize O1 and O2, it does satisfy several of the above desiderata insofar as it decouples data collection and learning and optimizes data collection actively and separately from the task solutions. This is achieved as follows: (a) The agent optimizes for several auxiliary objectives in parallel to the policies for the primary tasks of interest. (b) This allows the agent to learn a set of auxiliary policies that can facilitate learning of one or more main tasks. (c) These auxiliary policies are deployed to collect better experience. (d) Knowledge is shared across tasks by sharing experience. (e) Execution of auxiliary policies is actively scheduled to improve data collection for the main task. (f) This process is optimized via a separate learning process. The use of auxiliary policies bears some similarity to the role of skills in hierarchical architectures but there are two important differences: (1) Unlike skills, auxiliary policies are not directly used as part of the solution for the main task. The task policy is learned off-policy from the data collected with the auxiliary policies. (2) Execution of the auxiliary policies is scheduled to improve data collection. Although SAC-X emphasizes knowledge

sharing via data, as discussed in e.g. [22], this can be flexibly combined with a direct reuse of learned behavior representations such as skills.

## 3 A formal look at Collect and Infer

We provide a partial formalization of the ideas introduced in Section 2. We consider the standard objective consisting of an agent characterized by policy $\pi(a|s)$ acting in an environment $\mathcal{E}$ with states $s \in \mathcal{S}$, actions $a \in \mathcal{A}$, transition probability distribution $p(s_{t+1}|s_t, a_t)$, initial state distribution $p(s_0)$, and reward function $r$. The goal is to find a policy maximizing the expected sum of rewards

$$J(\pi_\theta) = \mathbb{E}_{\tau \sim \pi_\theta} \left[ \sum_{t=0}^{T} r(s_t) \right], \tag{1}$$

where $\tau = [(s_0, a_0), (s_1, a_1), \dots]$ is a trajectory of length $T$ sampled according to $p$ and $\pi$.

The main perspective change of *C&I* is that inference of the policy happens through optimization of a 'surrogate objective' defined in terms of a finite set of data. As a result, the optimization of the data set itself becomes part of the learning process. Thus, *C&I* can be characterized in terms of two operators: a) an 'Inference' operator, $\mathcal{I}$, that given a data set $\mathcal{D}$, computes a policy $\pi_\theta(a|s)$ and b) a data generation operator, $\mathcal{C}$, that generates the data set $\mathcal{D}$.

More precisely, the collection operator $\mathcal{C}$ will generate a data set consisting of $N$ trajectories, for instance by executing a collection policy $\mu$ in $\mathcal{E}$: $\mathcal{D}_c = \{\tau^1, \dots, \tau^N | \tau^i \sim \mu, p\} = \mathcal{C}(\mathcal{E}, N)$. The inference operator $\mathcal{I}$ optimizes $\pi$ to find the maximum of the surrogate objective $\mathcal{L}_I$ which is defined in terms of data $\mathcal{D}_c$: $\pi_\theta = \mathcal{I}(\mathcal{D}_c) = \arg\max_{\pi_\theta} \mathcal{L}_I(\pi_\theta, \mathcal{D}_c)$, This allows us to express a joint objective that couples (O1) – identifying an optimal policy given fixed data – and (O2) – identifying an optimal collection process given an inference procedure:

$$\mathcal{O}(\mathcal{C}; \mathcal{I}, N) = J(\arg\max_{\pi_\theta} \mathcal{L}_{\mathcal{I}}(\pi_\theta, \mathcal{D}_c = \mathcal{C}(\mathcal{E}, N))). \tag{2}$$

We measure the success of the policy inferred from the data collected by $\mathcal{C}$. For any choice of $\mathcal{I}$, environment $\mathcal{E}$, and fixed data budget $N$ we can identify an optimal collection process via the 'outer' optimization $\mathcal{C}^* = \arg\max_{\mathcal{C}} \mathcal{O}(\mathcal{I}, \mathcal{C}, N)$, for instance by optimizing $\mu$. Different choices for $\mathcal{I}$ will lead to different algorithms with different requirements for $\mathcal{C}$. For instance, we can obtain an algorithm in which we first create a fixed dataset and then obtain the policy via offline RL.

In practice, in particular the optimization with respect to $\mathcal{C}$ may be intractable and heuristics may be used instead. Furthermore, the collection process and the inference process may be tightly coupled and proceed in an iterative scheme. For example, the collection process might depend on previous estimates of an optimal policy (or previous data).

## 4 *Collect & Infer* and the state-of-the art in reinforcement learning

The example in Section 2 highlights that the *C&I* paradigm offers considerable flexibility. It suggests an interpolation between pure offline (batch) and more conventional online learning scenarios, and thus chimes naturally with the growing interest in data driven approaches, where large datasets of experience are built up over time, which can then enable rapid learning of new behaviors with only small amounts of online experience. Decoupling acting and learning, and the emphasis on off-policy learning gives greater flexibility when designing exploration or other actively optimized data collection strategies, including schemes for unsupervised RL and unsupervised skill discovery. Considering data as a vehicle for knowledge transfer enables new algorithms for multi-task and transfer scenarios. It finally suggests a different emphasis when thinking about meta-learning or life-long learning scenarios. To enable rapid adaptation to a novel task we may, for instance, focus on collecting a dataset that is suitable for learning new tasks offline, relying only on small amounts of task-specific online experience [e.g. 23, 24, 16, 25]. And in a similar vein we may use historic experience to mitigate problems associated with catastrophic forgetting [e.g. 26]. Many of these ideas are already present in the literature. However, we believe that embracing the versatility of off-policy learning and a stricter separation between data collection and inference will lead to future gains:

Model-free off-policy algorithms have improved considerably and are now widely used [e.g 9, 10, 11, 27]. However, they often continue to operate in an online fashion, without a clear separation of policy optimization and data collection. A more recent development are specialized algorithms that successfully operate in fully offline settings where a policy is optimized from a fixed dataset without

further interaction with the environment [15, 16, 18, 28]. Moving forward it will be important to focus on algorithms that work well in *both* the online and the offline setting [e.g. 29, 30] and can, for instance, combine large, stored 'offline' datasets and smaller amounts of 'online' experience.

The separation between the behavior that is executed and behavior that is optimized during learning has been exploited in goal-conditional or multi-task settings [e.g. 31, 32] and hierarchical goal-conditional [e.g. 33, 34] settings. More generally, there is growing interest in off-policy HRL algorithms [e.g. 33, 35, 36, 37, 22, 38] and offline skill-learning architectures [e.g. 39, 40, 41, 42, 43, 44]. So far, online and offline skill-learning architectures have, however, remained largely disjoint, and skills tend to be reused as part of a hierarchical target policy. The *C&I* paradigm encourages further integration of online- and offline skill learning architectures, a shift from the use of skills for policy optimization towards data collection, and more generally novel synergies between transfer via experience (data) and via parameterized representations [e.g. 24, 16, 25, 43]. This perspective also naturally integrates the use of expert demonstrations (data) or controllers (policies) during learning.

Similar to skills, stored trajectory data can be used to learn dynamics models for model based policy optimization [e.g. 9, 45, 46, 47]. The *C&I* paradigm suggests instead, to use such models for online behavior optimization during data collection. This can have the benefit that model error does not directly affect the learned policy [e.g. 48, 49, 50, 51], and that behavior can adapt rapidly, e.g. to alternative rewards. More generally, the *C&I* paradigm naturally allows for multiple behavior rules with different levels of amortization/'on-the-fly' optimization (see also e.g. [29, 52]).

The *C&I* viewpoint emphasizes the optimization of data collection. While a fully Bayesian treatment can provide an optimal trade-off between exploration and exploitation [53, 54] but it is usually intractable. Various alternative objectives have been explored both in the supervised and unsupervised setting. These include approximate treatments of uncertainty [e.g. 55, 56, 57], intrinsic rewards derived from sensor changes [e.g. 58, 59], motivated by empowerment or related information-theoretic formulations [e.g. 60, 61, 62, 63] and curiosity-based objectives [e.g. 64, 65, 66, 67]. The *C&I* model encourages further research into strategies for the acquisition of information it provides a flexible framework that may facilitate a conceptual disentanglement of objectives, representations, and execution strategies. Information acquisition strategies could be motivated by, or explicitly employ, Bayesian reasoning, but the framework does not narrowly prescribe this perspective. However, separating data collection and processing disentangles the evidence from the inferred quantities.

## 5    Conclusions and outlook

*C&I*-paradigm aims to re-think data-efficient RL through a clear separation of data collection and exploitation into two distinct but connected processes and to exploit the flexibility of off-policy RL in agent design for problems as diverse as online RL, offline RL, or lifelong-learning. We hope this will inspire and intensify a several research avenues, of which we just want to highlight a few:

- An optimal collect process in the sense of O2 is key for data-efficient agents and thus for achieving AI. This requires awareness of the knowledge that the agent has already acquired. A dedicated research agenda should consider: What is a good objective for collecting the 'right' data? What surrogates could we use if the 'correct' objective is impractical?

- Effective implementations of 'infer', i.e. how to squeeze all of the knowledge out of existing data; how to learn efficiently and reliably from large existing datasets, and how to optimally merge the on-line and the off-line viewpoint on data generation and exploitation?

- *C&I* emphasizes the reuse of previously collected experience. This raises the question what other intermediate representations of knowledge, besides policies, can be extracted from data and efficiently reintegrated into the process of data collection and inference (e.g. skills, models, rewards) to improve the capabilities of the learning system?

*C&I* is not tailored to a particular learning scenario and its applications range from 'classical' single task learning scenarios to multi-task scenarios. Going forward, we see *C&I* as a natural basis for a data-efficient learning agent, that treats data as a raw resource that can be flexibly transformed into different types of representations that can be used, for instance, for action selection (e.g. policies), or may facilitate future learning problems (e.g. models, perceptual representations, or skills). At any given time, the agent may act to collect new data, either with the goal of improving its performance on a particular, external task, or to simply learn more about its environment in a way that can be exploited in the future.

**Acknowledgments**

Thanks to the Control Team and various colleagues at DeepMind for ongoing discussions, and in particular to Patrick Pilarski, Dan Mankowitz and Markus Wulfmeier for their valuable comments on the manuscript.

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
