# OpenReview forum: "Collect & Infer - a fresh look at data-efficient Reinforcement Learning"
_robot-learning.org/CoRL/2021/Conference/Blue_Sky — CoRL 2021, Blue Sky_

### Official Review · Reviewer_XZya · 2021-08-25

**Novelty:** Good
**Impact:** 4
**Clarity Of Presentation:** Very Good

**Recommendation:**

Strong Accept: I recommend accepting the paper and will argue for my recommendation even if other reviewers hold a different opinion.

**Summary:**

The paper presents the Collect and Infer (C&I) paradigm, which clearly separates the data collection and inference processes in reinforcement learning. The paper first describes the core idea of the C&I paradigm and its implications. Then, the formulation of the C&I paradigm is presented. Finally, the connection to recent studies are discussed.

**Summary Of Recommendation:**

I enjoyed reading the paper. The C&I paradigm provides a new perspective on RL. In addition, the C&I paradigm shed light on the importance of optimizing the data collection process, which is often overlooked. The paper provides a new perspective on RL, and the C&I paradigm presented in the paper is worth being shared at CoRL. The paper indicated by that datasets such as D4RL accelerate the research on the inference process, but a research framework that accelerates the research on the data collection process should also be developed.

I have some minor questions.
-	In the abstract, the term “transition memory based RL” seems equivalent to “offline RL.” Can you clarify why the term “transition memory based RL” is used in the abstract?

-	I think that the C&I paradigm presented in the paper will also be a key for bringing the imitation learning and RL. Can you comment on how imitation learning can be connected to the C&I paradigm?

---

### Official Review · Reviewer_qPwV · 2021-09-03

**Novelty:** Good
**Impact:** 4
**Clarity Of Presentation:** Excellent

**Recommendation:**

Strong Accept: I recommend accepting the paper and will argue for my recommendation even if other reviewers hold a different opinion.

**Summary:**

This is a position paper which advocates for the "collect and infer" (C&I) framework whereby the experience collection (collect) and policy inference (infer) aspects of reinforcement learning are explicitly separated and optimized. Classic tabular RL methods collect experiences and learn in parallel. In contrast, the C&I agent would 1) do batch policy optimization given a fixed experience buffer; 2) collect experiences that optimize the expected quality of the policy learned in #1. The hope is that this approach would enable us to define the problem of experience collection (i.e. exploration in RL) as an optimization problem with respect to a fixed policy optimization strategy.

I found this paper easy to understand and the overall proposal compelling. I like the simplicity of the idea and I like that it seems to provide a different framework from which to think about exploration. There were strong connections to recent literature in the field and a long reference list. While reading this, I wondered about what the connection might be to Bayes adaptive RL. In principle, that would be the optimal approach to exploration. So, is C&I an approximation or is there no relationship at all? It seems like there are some assumptions embedded in the approach of eq 2 which are unstated.



**Summary Of Recommendation:**

I thought this was an interesting position paper. It's not completely novel as the C&I idea seems to be percolating in the literature, but I thought it offered great perspective on the idea.

---

### Decision · Program_Chairs · 2021-10-01

**Decision:**

Accept

**Comment:**

All reviewers found the paper to be an interesting position paper on data reusage in deep RL.